# Self-sustained electricity generator driven by the compatible integration of ambient moisture adsorption and evaporation

Jin Tan[1], Sunmiao Fang[1], Zhuhua Zhang [1,2], Jun Yin [1], Luxian Li[1], Xiang Wang[1] & Wanlin Guo [1,2 ✉]

Generating sustainable electricity from ambient humidity and natural evaporation has attracted tremendous interest recently as it requires no extra mechanical energy input and is deployable across all weather and geography conditions. Here, we present a device prototype for enhanced power generation from ambient humidity. This prototype uses both heterogenous materials assembled from a LiCl-loaded cellulon paper to facilitate moisture adsorption and a carbon-black-loaded cellulon paper to promote water evaporation. Exposing such a centimeter-sized device to ambient humidity can produce voltages of around 0.78 V and a current of around 7.5 μA, both of which can be sustained for more than 10 days. The enhanced electric output and durability are due to the continuous water flow that is directed by evaporation through numerous, negatively charged channels within the cellulon papers. The voltage and current exhibit an excellent scaling behavior upon device integration to sufficiently power commercial devices including even cell phones. The results open a promising prospect of sustainable electricity generation based on a synergy between spontaneous moisture adsorption and water evaporation.

---

[1] Key Laboratory for Intelligent Nano Materials and Devices of the Ministry of Education, State Key Laboratory of Mechanics and Control of Mechanical Structures, Nanjing University of Aeronautics and Astronautics, 210016 Nanjing, China. [2] Institute for Frontier Science, Nanjing University of Aeronautics and Astronautics, 210016 Nanjing, China. ✉email: wlguo@nuaa.edu.cn

Recently emerged hydrovoltaic technologies can harvest electricity from the ubiquitous water available in nature, thus opening up a promising source of green energy[1–3]. These new technologies generate electricity directly from the electric coupling at interfaces between materials and various forms of water[4], including moisture[5–7], raindrops[8,9], and waves[10]. Such solid–liquid coupling demonstrates a superior technology for its high flexibility and eco-friendly development[11–13]. Especially, the evaporation- and moisture-induced electricity stands out because they can operate independently in a wide range of environments with no need of mechanical input[14–16]. Evaporation generators can convert thermal energy from the air into sustainable electricity until water is completely vaporized out[17–20], while humidity-driven generators directly absorb moisture in the ambient to generate pulse electric signals[21,22]. Both represent promising ways of spontaneous power generation but have their own limitations from water source demand and unsustainable electricity output, respectively. It is in urgent need to explore new patterns for generating continuous electricity from ambient environments. Recently, exciting progress in this domain has been made in polyelectrolyte films and biological nanowires[23,24], high-voltage output for long duration was obtained while the current output can hardly be sustained as long without attenuation, which heavily inhibit their practical applications. Integrating moisture adsorption and water evaporation to establish an intact and persistent water cycle in one device provides an ideal means for achieving persistent electricity generation[25]. Yet, it remains a great challenge in combining the advantages of ambient humidity and evaporation since a high ambient humidity often slows down the evaporation.

Pursuing this concept, we present here a self-sustained electricity generator (SSEG) driven by the integration of moisture adsorption with water evaporation. The SSEG consists of a hygroscopic layer and an evaporative layer. The two layers are made from cellulon papers that are decorated with different additives in each case. The assembled device can absorb moisture from the air through its hygroscopic layer and, simultaneously, vaporize water out through the evaporative layer, thus generating a persistent running water circle. Because of the uninterrupted water flow through the negatively charged channels—flow driven by directed evaporation—the SSEG is able spontaneously to produce both a continuous voltage of ~0.78 V and a continuous current of ~7.5 µA, both of which last for more than 10 days in an ambient environment. In addition, the SSEG exhibits favorable reliability in a wide range of operating temperatures (−5 °C to 35 °C) and relative humidity (RH, 20–80%). By series/parallel connections of multiple generators, the power outputs can be scaled up easily, which is enough to directly power commercial electronics. This study offers a strategy for durable electricity generation and paves the way for designing an efficient, reliable generator for practical applications that can run across a range of ambient environments.

## Results

**Self-sustained electricity generation of the SSEG.** As shown in Fig. 1a, the SSEG consists of a hygroscopic layer and an evaporative layer with two carbon tape electrodes connected to their external surfaces. The hygroscopic layer was fabricated by introducing LiCl into a cellulon paper slice in a simple impregnation method accompanied with further oxygen plasma treatment (see details in experimental section). As for the evaporative layer, hydrophobic carbon black was doped into another cellulon paper slice to form an asymmetrical distribution in the vertical direction. Due to the highly porous structure and fluffy multi-layered architecture of the cellulon paper slice, LiCl and carbon

black effectively attach to the surface and enter the inner porous fibers (Fig. 1b and Supplementary Fig. 1)[26]. The surface of the hygroscopic layer presents a super hydrophilicity due to the large number of pores and hydrophilic groups[27,28]. On the contrary, the outside surface of the evaporative layer with an abundant distribution of hydrophobic carbon black exhibits a large water contact angle of 116.48°, even as the inner surface is partially hydrophilic because of the relatively small amount of carbon black (Fig. 1b). The conspicuous diversity of hydrophilicity and hydrophobicity endows the hygroscopic layer and the evaporative layer the distinct ability to pump moisture and exhale vapor. This gives rise to the continuous transport of water and ions across the bilayer structure[29,30]. In a general ambient environment of 25 ± 2 °C and RH 60 ± 5%, the square SSEG with a side length of 3 cm, unless otherwise claimed, can generate a continuous electric output of about 0.78 V and 7.5 µA for a duration of more than 10 days (Fig. 1c). The duration of the current output presents a more than tenfold improvement over the results from previous ambient generators with a single process of moisture adsorption[23,31]. The maximum power density of 0.7 µW cm$^{-2}$ as well as gravimetric energy density of 0.15 mW h kg$^{-1}$ were measured and calculated (Supplementary Fig. 2). The voltage and current vary with load resistances, and a maximum volumetric power output of 0.67 µW cm$^{-3}$ can be achieved at an optimal resistance of ~100 kΩ (Supplementary Fig. 3). Removal of oxygen from the gas phase has little impact on electricity generation (Supplementary Fig. 4). When the carbon tape electrodes were replaced by two gold electrodes, the device has similar electric outputs, indicating that electrodes do not make contributions to the observed electrical outputs (Supplementary Fig. 5).

**Moisture adsorption and water evaporation behaviors of the SSEG.** After ruling out the contribution of an activating reaction caused by oxygen and electrodes, we infer that the generation of the continuous electrical output of the SSEG could be attributed to the persistent water flow produced by the compatible integration of moisture adsorption and water evaporation. Thus, the specific hygroscopic capacity and evaporation behavior of the SSEG were gravimetrically evaluated with an analytical balance at 25 ± 3 °C. When the hygroscopic layer is exposed to air, the lithium chloride contained inside quickly absorbs moisture from the air—within half an hour—through spontaneous deliquescence (Supplementary Fig. 6). Benefitting from the super hydrophilicity and the porous nanofibrous structure of the cellulon paper (Fig. 1b and Supplementary Fig. 7), moisture adsorption and water transfer rates of the hygroscopic layer are remarkable. The absorbed water content of the hygroscopic layer increases within two hours and reaches saturation after approximately nine hours (Fig. 2a). The moisture absorption capacities of the dried hygroscopic layer at 60%, 70%, and 80% RH attain 0.43, 0.75, and 1.14 g/g, respectively. Then, we assembled an evaporative layer over a dried hygroscopic layer and measured the hygroscopic capacity of the bilayer structures in the same condition. After adding the evaporative layer, the moisture adsorption capacities at 60%, 70%, and 80% RH reduce to 0.28, 0.38, 0.53 g/g, respectively. The weight of the device increases rapidly and then reaches a plateau of about 130% the initial weight at 25 °C, 70% RH (Supplementary Fig. 8), the time to reach the equilibrium of water content increases from 9 h to over 15 h (Fig. 2b). The results could be attributed to the mass transfer process from the hygroscopic layer to the evaporative layer. As shown in Fig. 2c, d, the mass change of the separated hygroscopic layer (I) reaches 0.77 g/g at 25 °C and 70% RH (baseline). When there is a mass transfer process between the two layers in devices 1 and 2, the mass change in the hygroscopic layer reduces to 0.59 and 0.54 g/g

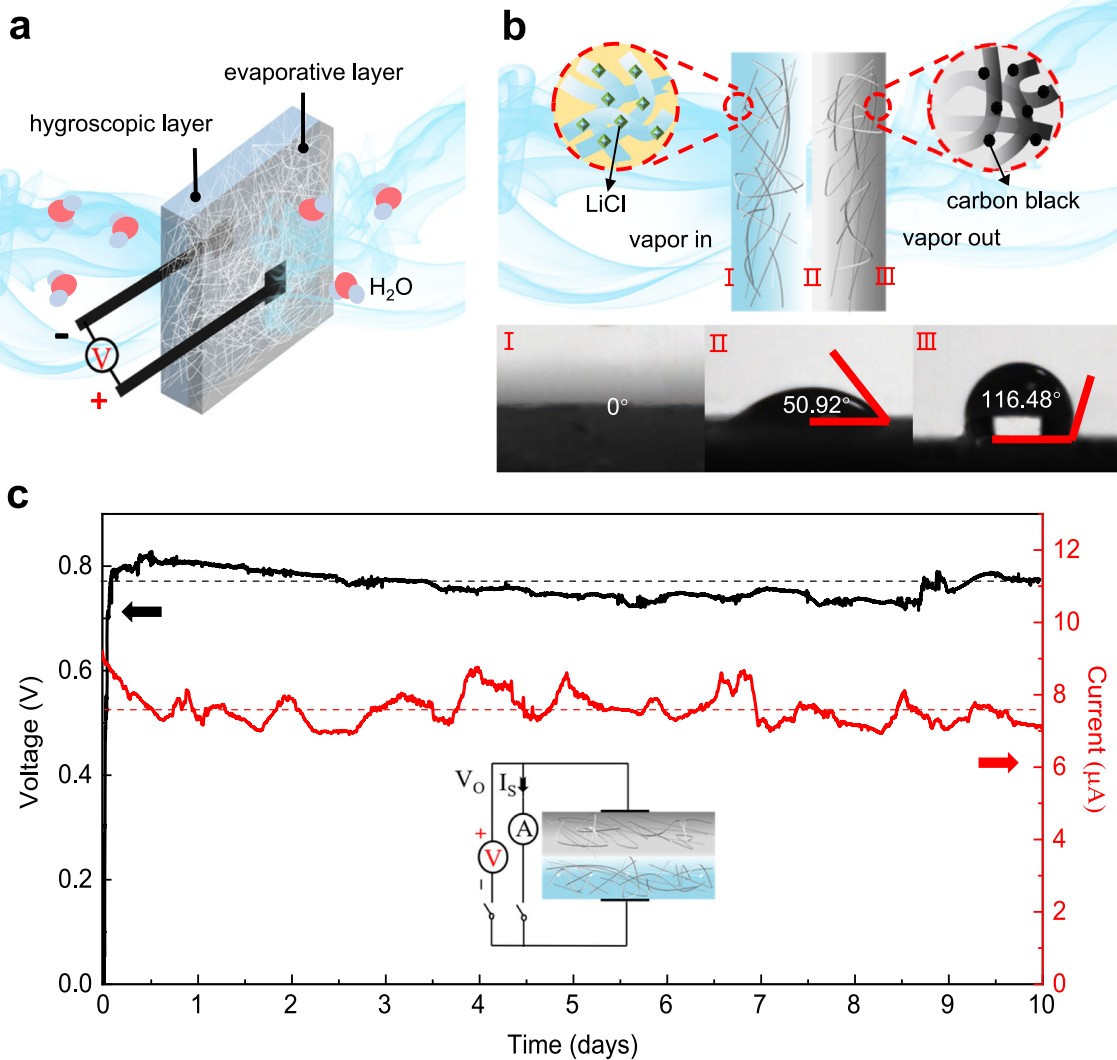

**Fig. 1 Schematic and electric output characteristics of the SSEG. a** Schematic of the bilayer SSEG. The bilayer SSEG consists of a hygroscopic layer and an evaporative layer with two carbon tape electrodes placed on their external surfaces. **b** Diagram of the device structure from the side. Water contact angles of the hygroscopic layer (I), the inner side (II) and the outer side (III) of the evaporative layer are demonstrated at the bottom. **c** A continuous recording of open-circuit voltage (black) and short-circuit current output (red) from the SSEG for ten days in an ambient environment (25 ± 2 °C, RH 60 ± 5%). The black and red dashed lines represent the average value of the open-circuit voltage and the short-circuit current output, respectively. The inset is the corresponding circuit model.

(III and V), respectively. The mass transfer also occurs in the evaporative layer, although much weaker. Compared to the near-zero mass change of the separated evaporative layer (II, baseline), the mass change of the evaporative layers in devices 1 and 2 (IV and VI) reaches up to 0.21 and 0.16 g/g, respectively. The relatively lower mass change in the evaporative layer in device 2 can be attributed to the natural evaporation of water into the air. The water absorptivity of the evaporative layer was also evaluated at different RH. It is found that 23.7%, 29.3%, and 31.8% of the water adsorbed by the hygroscopic layer flows to the evaporative layer at 60%, 70%, and 80% RH, respectively (Supplementary Fig. 9). In addition, LiCl will not be massively consumed and crystallized at the evaporative layer during the mass transfer process due to the high restriction to chloride-ion-flow (Supplementary Fig. 10). On account of the difference of mass change in the hygroscopic layer and the evaporative layer, a water content gradient forms between the two layers, as shown directly by the water content map of the device (Fig. 2e). From the external surface of the hygroscopic layer to the external surface of the

evaporative layer, the water content of the seven layers decreases from 0.35 to 0.04 g/g, verifying the presence of the obvious water content gradient inside the bilayer device.

To further clarify the water evaporation behavior of the SSEG, illumination was applied onto an integrated bilayer structure with a sealed and water-filled hygroscopic layer. Compared to the bilayer hygroscopic layer with cellulon paper and a mass loss of the hygroscopic layer, the evaporative layer presents a more than twofold increase under one-sun illumination (Fig. 2f), indicating powerful evaporation facilitation brought by the evaporative layer. The enhanced evaporation rate could be attributed to the highly open 3D nanofiber networks and the hydrophobic outside surface of the carbon-black loaded evaporative layer. In addition, the heat generated by the efficient absorption of broadband solar radiation makes the contribution occur at the same time (Supplementary Figs. 9 and 11). It is also shown that heat transfer between the two layers will not reduce the electric output of the device (Supplementary Fig. 12). Through the synergistic effect of the hygroscopic and evaporative layers, the designed

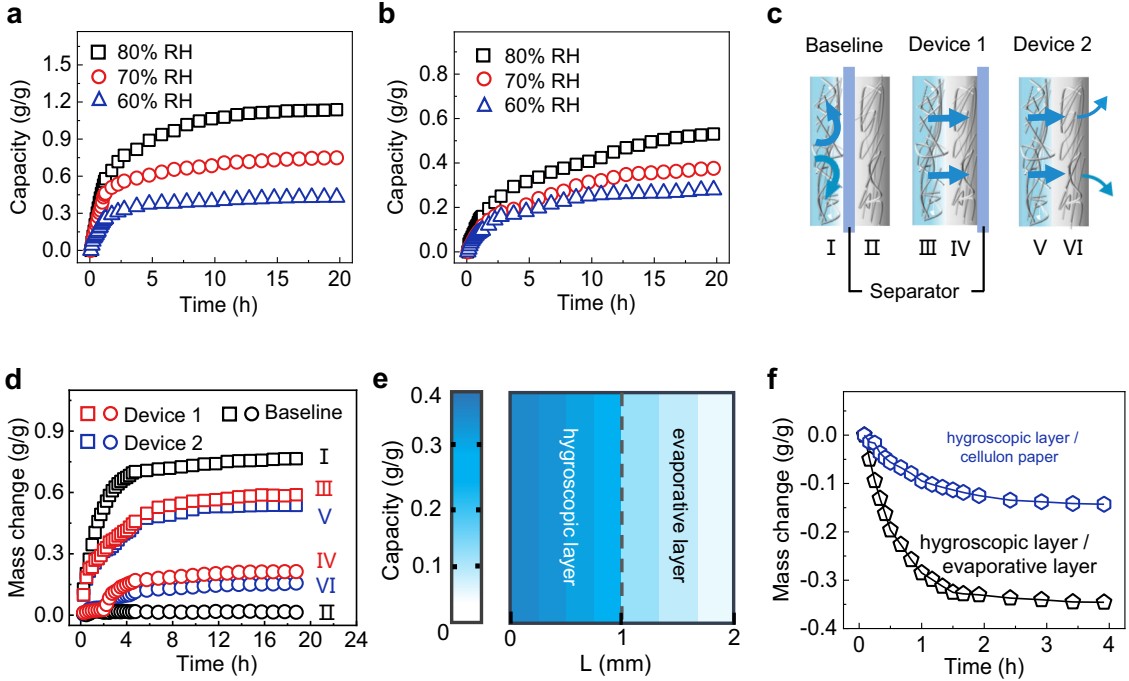

**Fig. 2 Moisture adsorption and water evaporation behaviors of the hygroscopic layer and the evaporative layer. a**, **b** Moisture adsorption kinetics of the hygroscopic layer and the hygroscopic/ evaporative bilayer structure. **c** Schematic of mass transfer between the two layers. Baseline: The hygroscopic layer and the evaporative layer was separated by a separator to prevent mass transfer between them. Device 1: The outside surface of the evaporative layer was covered by a separator to prevent evaporation. Device 2: Without any additional refitment. The letters I-VI represent the hygroscopic layer and the evaporative layer of the three devices, respectively. The arrows represent the transfer of water. **d** Mass change in different parts of the devices at 25 °C, 70% RH. **e** Water content map of the SSEG after moisture adsorption in 60% RH for over 30 h. The water content map was depicted by gravimetrically analyzing water absorbed by each layer. The resolution corresponds to the film thickness, which is 0.25 mm for the hygroscopic layer, and 0.33 mm for the evaporative layer. **f** Mass change in the bilayer structures of the hygroscopic layer/cellulon paper and the hygroscopic layer/evaporative layer versus time under one-sun illumination.

bilayer structure can simultaneously absorb moisture and evaporate water, thus generating a water content gradient and maintaining the water transportation inside the bilayer device.

**Mechanism of electricity generation in the SSEG.** In order to reveal the electricity generation mechanism of the SSEG, we conducted a series of control experiments adjusting the balance between moisture adsorption and water evaporation. Figure 3a presents the voltage output of a device with both sealed hygroscopic and evaporative layers. When the device was fully sealed with a parafilm, the voltage produced was near zero, indicating that electricity generation is closely related to the presence of water in the devices. After the parafilm encapsulated on the evaporative layer was removed and the device was exposed to an environment of 25 °C and 60% RH. The voltage output remained at a stable level of nearly zero and the device remained dry even after several hours (Fig. 3b), which demonstrates that moisture can hardly be absorbed by the hygroscopic layer through the evaporative layer. We then removed all the sealed parafilm so that the whole device was exposed to the air. The lithium chloride attached to the cellulon paper started to deliquesce and then absorbed moisture from the air, forming a water content gradient and directional water flow inside the device, as aforementioned. The SSEG without any seal produces a continuous electrical output of 0.8 V, 7 μA (Supplementary Fig. 13), suggesting that the water content gradient and directional transport of water are crucial to electricity generation. After operating for several hours, the evaporative side of the device was sealed, evaporation stops and water gradually reaches saturation in the bilayer structure. As a result, the electric output progressively declines from 0.68 V to

less than 0.1 V in about 5 h (Fig. 3c). Besides, blocking the moisture adsorption and evaporation of an operating device at the same time also leads to serious degradation of electric output (Supplementary Fig. 14). These results provide strong evidence that the continuous and directional water flow driven by evaporation plays a vital role in electricity generation in the bilayer device.

Zeta potential measurement and the X-ray photoelectron spectroscopy characterization show that cellulon fibers treated by plasma and doping carbon black display negatively charges, which are resulted from the abundant oxygen-containing functional groups on the surface (Fig. 3d and Supplementary Fig. 15). Once ionic solutions flow through the negatively charged channels in the cellulon paper, the channels should exhibit a high selectivity to pass through cations and repel anions[7,18,32,33]. Therefore, the detailed mechanism of electricity generation could be proposed, as illustrated schematically in Fig. 3e. When the SSEG is exposed to air, the LiCl contained in the hygroscopic layer absorbs moisture rapidly and is partially dissociated into ions, thus forming a water content gradient and ion concentration gradient inside the bilayer structures. As a result, water and ions, including protons and lithium ions, will flow directionally through the channels and micropores from the hygroscopic layer to the evaporative layer (Supplementary Fig. 16), thus inducing voltage and current outputs due to evaporation-induced electricity and ion movement[5,14,31]. The proposed mechanism can be further supported by the electric output contributed by different layers (Fig. 3f), in which the hygroscopic layer with only ion movement (protons and lithium ions) yields a voltage of 0.22 V ($V_{1-2}$), while the evaporative layer with both evaporation and ion

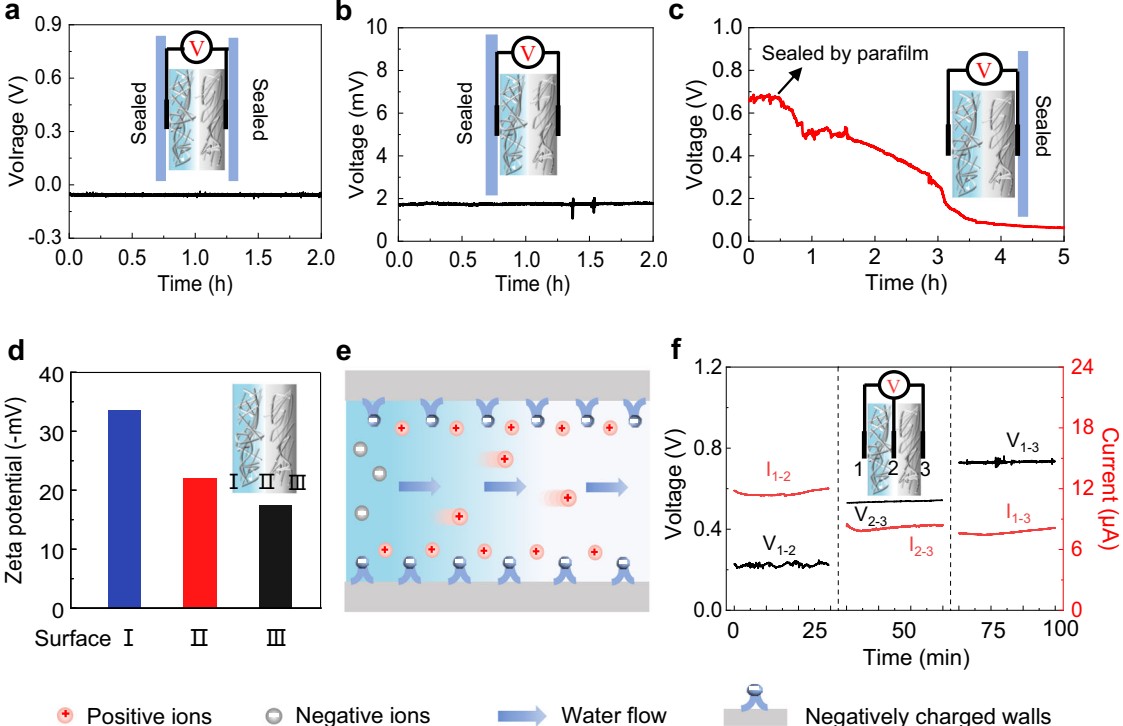

**Fig. 3 Mechanism for electric output of the SSEG. a** Voltage signal produced by the device with both the hygroscopic layer and evaporative layer sealed. **b** Voltage signal produced by the device with only the hygroscopic layer sealed. **c** A diminishing voltage signal produced by the device with only the evaporative layer sealed. **d** Zeta potential of the hygroscopic layer (I), the inner side (II), and the outer side (III) of the evaporative layer. **e** Schematic illustration of the working mechanism of the SSEG. **f** Voltage and current outputs of different component layers at 25 °C and 60% RH. The internal resistances of the hygroscopic layer and the evaporative layer are ~16.2 kΩ and ~71.4 kΩ, respectively. The numbers 1–3 represent the electrodes in different positions.

movement yields an obviously higher voltage of 0.51 V ($V_{2-3}$). In previous ambient environment generators with a single process of absorbing moisture, the current output in a closed circuit will rapidly decline due to the unsustainable water content gradient[12,22]. By harvesting energy from the environment by integrating moisture adsorption with water evaporation (Supplementary Fig. 17), the water content gradient inside the SSEG could sustain for a long duration and induce a persistent driving force for directed water flow and ion movement beyond the back-migration driven by electrostatic repulsion[5,24], thus resulting in a timely self-charging process and enhanced duration of current output.

The role of lithium chloride during electricity generation was further explored by fabricating SSEGs with different hygroscopic agents. As shown in Supplementary Fig. 18, SSEGs with three kinds of nonionic hygroscopic agents, including metal-organic framework (UIO-66), zeolite, and clay, can yield voltage outputs of about 0.4, 0.25, and 0.2 V, respectively. These voltage outputs are in the same order of magnitude but slightly lower than that of the SSEG using lithium chloride as hygroscopic agent. Considering the lower hygroscopic capacity of the nonionic hygroscopic agents[34], the results indicate that sustained water evaporation and proton transfer play a more important role than lithium-ion movement in electricity generation. The SSEGs with other deliquescence salts as hygroscopic agents, including $MgCl_2$, $CaCl_2$, $AlCl_3$, and $FeCl_3$, also yield electric outputs as shown in Fig. 4d, which will be discussed in detail later.

**Electric outputs performance of SSEGs with different compositions and under different conditions.** Further investigation demonstrated that the electric output performance of the SSEG is closely bound up with the external environmental conditions including ambient temperature and humidity. Voltage increased from 0.3 V to over 0.7 V when the temperature increased from −5 °C to 25 °C. As the temperature further rises to 35 °C, the voltage output exhibited a slight decrease of 0.02 V (Fig. 4a). The voltage variation as humidity increases from 20% RH to 80% RH exhibits a similar tendency that climbs up first and then declines, with a maximum value of 0.78 V at 60% RH (Fig. 4b). The results can be attributed to the balance between moisture adsorption and water evaporation under different environmental conditions. Temperature and humidity play important roles in the processes of moisture adsorption and evaporation. High temperature and low humidity favor rapid evaporation of water, while high humidity is in favor of moisture adsorption. When these two parameters reach the appropriate values, a balance is established between pumping moisture and exhaling vapor. This balance is maintained for a long time, spawning the efficient directional transport of water molecules and ions within the device, thus generating a maximum electric output. The remarkable performance of the SSEG under a wide range of temperatures and degrees of humidity indicates great potential for practical applications.

The continuous water flow driven by the compatible integration of moisture adsorption with water evaporation plays a key role in electricity generation. The moisture capturing ability of the hygroscopic layer can greatly influence the performance of the SSEG. We further investigated the impact of the amount of LiCl contained in the hygroscopic layer. Since the LiCl was doped by impregnating a cellulon paper slice into the LiCl solution, we can control the content of LiCl by varying the concentration of the solution. With a continuous increase of the LiCl content from 5 to

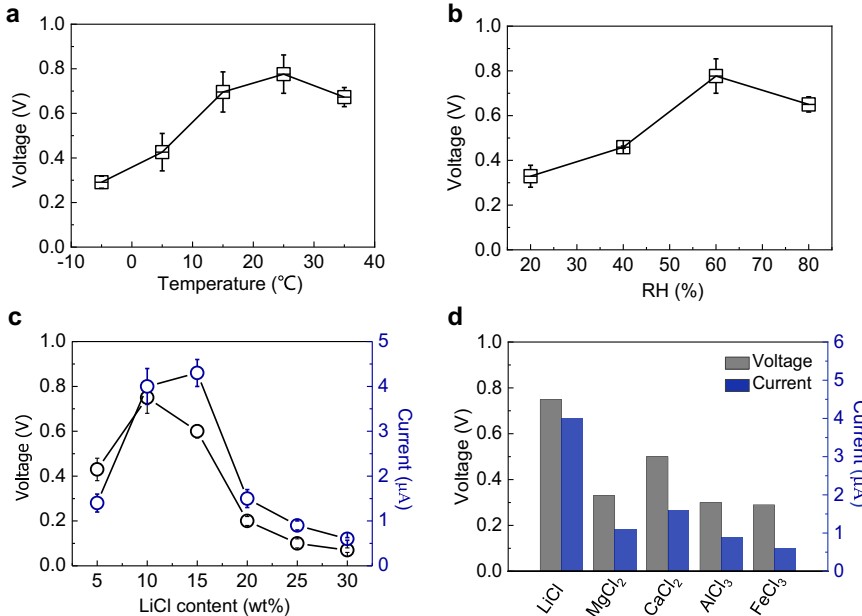

**Fig. 4 Electric output performance of the SSEG. a** Open-circuit voltage variation versus different temperatures at 50% RH. **b** Open-circuit voltage variation versus different RH at 25 °C. **c** Electric output values variation of the SSEG with an incremental LiCl content contained in the hygroscopic layer. **d** Measured open-circuit voltage and short-circuit current of the SSEG with various hygroscopic salts contained in the hygroscopic layer. The vertical error bars represent the standard deviation of the measured peak values.

30%, the electric output of the device increases first, reaches its maximum value at a moderate content ~10%, and then decreases. When the LiCl content is too low, the moisture absorption ability of the hygroscopic layer will be not enough to replenish the water escaping from the evaporative layer, resulting in a scant amount of flowing water and ions. When the LiCl content is higher than 15%, a great deal of water could be absorbed. Water gradually filled the device, attenuating the electric output consequently (Fig. 4c and Supplementary Fig. 19). Furthermore, the introduction of LiCl can dramatically increase the ionic conductivity of the device to 1.5 µS/cm at 50% RH (Supplementary Fig. 20), reduce the resistive loss, and lead to a higher current output compared to devices without LiCl (Supplementary Fig. 21).

In addition to LiCl, four common hygroscopic salts including $MgCl_2$, $CaCl_2$, $AlCl_3$, and $FeCl_3$ were separately introduced into the hygroscopic layers at the same concentration[35–37]. Electric outputs of the obtained devices were then measured successively. As the ion movement makes contribution to electricity generation, factors including ionic radius and ionic valence that can affect ion diffusion behavior will also affect the electric output of the device. As shown in Fig. 4d, the SSEG with LiCl as the hygroscopic agent outputs the highest electricity, due to the effective ion diffusion with small ionic radius and low electrostatic repulsive force with low ionic valence. Other hygroscopic agents with larger ionic radius or higher ionic valence result in lower generation performance of the SSEG.

According to the proposed mechanism, constructing materials with negatively charged channels and abundant oxygen-contained functional groups are ideal substitutes of cellulon paper. Expectedly, controlled devices made from lignocellulose aerogel and polyvinyl alcohol (PVA) hydrogel, have these two characteristics[38], can yield similar and even higher electric output of 0.9 V, 5.5 µA, and 0.5 V, 4 µA (Supplementary Fig. 22), respectively, supporting the proposed mechanism above. And a gravimetric energy density of 4.7 mW h kg$^{-1}$ is obtained by the device made from lignocellulose aerogel at 55% RH and 23 °C, ten times higher than previously reported moist-electric generators under similar condition[23,31].

The thickness of the device may affect the electricity generation performance as it is correlated to the ion diffusion distance and the gradient of water content. A SSEG with a thickness of 2 mm yields an optimum voltage and current output of 0.78 V and 7.5 µA, respectively (Supplementary Fig. 23). Furthermore, the SSEG maintained good performance retentions of 85% and 88% after autonomous charge–discharge for 150 cycles and periodic water adsorption–dehydration for 10 cycles, showing satisfactory cyclic stability under dynamic operation conditions (Supplementary Fig. 24).

**Scaling up and demonstration of SSEG.** The electric output of the SSEG can be scaled up by size regulation and multi-device connection. When the device size increases from 1 cm$^2$ to 9 cm$^2$, with the side length of the square device increasing from 1 to 3 cm, the current output increases from less than 1–4.3 µA, with the voltage remaining relatively stable at 0.7 V (Fig. 5a). Since the size increase can be regarded as a parallel connection of multiple elemental devices, the result presents a promising option for preparing large-area devices. The output voltage and current can be further boosted by series and parallel connections of multiple devices, respectively. Six devices that are series-connected, as illustrated by the circuit inset in Fig. 5b, are able to produce a continuous electric output of ~2 V and 10 µA. Five generators connected in a series generate a voltage of 2.9 V and can power an LCD screen as shown in Fig. 5c.

Integrating SSEG units on a large scale was then carried out. A hundred devices with series of parallel connections produced a continuous electric output of over 5 V and 40 µA (Supplementary Fig. 25). This output can directly light 5 LEDs (Fig. 5d and Supplementary Movie 1), in contrast to previously reported generators that need extra capacitors and rectifiers to light one LED[24]. The electric power produced by the SSEG can be stored in commercial energy-storage devices with no need of extra rectifiers and power management circuit, which greatly reduce the energy consumption compared to devices with unstable electric output. Capacitors of 100, 470, 1000, and 2200 µF could be charged to 5 V by the SSEG integration in several minutes and maintain a full

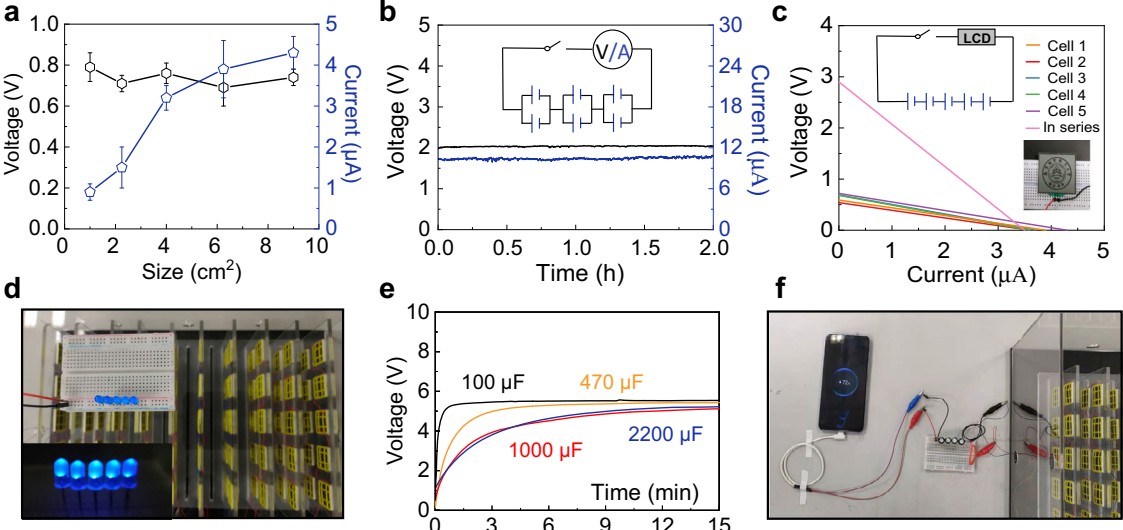

**Fig. 5 Integration and applications of the SSEG. a** Electric outputs plotted against the size of square device in an ambient environment of 25 °C, 60% RH. Error bars represent the standard deviation. **b** Electric outputs and circuit diagram of six individual devices with a series of parallel connections. The inset is the corresponding circuit model. **c** Current–voltage curves of five individual devices and their series connections. Photograph of an LCD screen (right lower set) driven by these five serial devices. The inset is the corresponding circuit model. **d** Photograph of five LEDs (left lower set) driven by 100 SSEGs with a series of parallel connections. **e** Voltage–time curves of commercial capacitors of varying capacitance (100, 470, 1000, and 2200 μF) charged by the integrated SSEGs. **f** A charging smartphone driven by the integrated SSEGs.

charge state (Fig. 5e), in contrast to previously reported integrated heterogeneous moist-electric generators that need a charging time of over ten hours to charge a capacity of 330 μF[5]. The accumulated electricity in ten minutes is demonstrated to successfully charge a cell phone for fifteen seconds and power Bluetooth devices (Fig. 5f and Supplementary Movies 2 and 3).

## Discussion

We have demonstrated a prototype for self-sustained electricity generators that are driven by integrating moisture adsorption and water evaporation. A device with an area of around 9 cm² could continuously generate both a remarkable voltage of about 0.78 V and a current of 7.5 μA for a duration of more than 10 days in the ambient environment. Compared with the previously reported methods of harvesting energy by absorbing moisture in ambient air, a sustainable water cycle and ceaseless directed water flow can form within the negatively charged channels in the device, resulting in the significantly enhanced duration of electricity output, which is mainly due to the electrokinetic effects. This integration strategy has been proven to work well with different constructing materials as well as nonionic hygroscopic agents. The device could work across a wide range of temperatures and relative humidity. The power outputs can be scaled up easily by interconnecting multiple devices and suffice to power commercial electronics. This desirable performance of electricity generation and environmental adaptability suggests that the SSEG has great potential for sustained energy harvesting in ambient environments.

## Methods

**Materials and reagents**. The hygroscopic salts including LiCl, CaCl₂, MgCl₂, FeCl₃, AlCl₃, the terpinelol, PVA ($M_w$ ~67,000), glutaraldehyde (50%wt in DI water), permutit ($M_w$ ~218) and montmorillonite were purchased from Aladdin Biochemical Technology Co., Ltd.. The glass microfibre membranes were purchased from GE Healthcare Life Sciences Co., Ltd. The nanofibrillated cellulose was purchased from Zhongshan nanofiber material Co., Ltd. The ethylcellulose was purchased from Shanghai Titan Scientific Co., Ltd. The toluene and hydrochloric acid were purchased from Nanjing Chemical Reagent Co., Ltd.

**Preparation of the hygroscopic layer**. The hydroscopic layer was prepared with a simple impregnation method. In total, 10 g lithium chloride was added into DI water (100 ml) and stirred for 30 min. Subsequently, a cellulon paper slice (3 cm × 3 cm × 1 mm) was immersed in the above solution for 30 min and then dried at 100 °C for 1 h. One side of the obtained cellulon paper slice loaded with LiCl was then modified with an oxygen plasma treatment for 5 min. The hygroscopic layers made from lignocellulose aerogel and fiberglass film were prepared in a similar impregnating and drying method. The PVA-based hygroscopic layer was prepared by adding LiCl solution (5 wt%) into the mixture of PVA and cross-linking agent before gelation happened. The hygroscopic layers with zeolite and clay as hygroscopic agents were prepared by repeating the impregnation and drying process several times until the hygroscopic agents were evenly distributed in the constructing materials. The hygroscopic layer with metal-organic framework (UIO-66) as hygroscopic agent was prepared by a solvothermal method with further washing and drying process.

**Preparation of the evaporative layer**. The carbon-black slurry was obtained by dispersing 1 g toluene carbon black, 2 g ethylcellulose, and 6 g terpinelol in 50 ml ethyl alcohol and then stirring for 3 h. A cellulon paper slice (3 cm × 3 cm × 1 mm) was immersed in the carbon-black slurry and then transferred to a hot plate at 100 °C to dry for 1 h. A piece of porous cloth was placed between the hot plate and the sample to induce drying and to realize a concentration gradient of carbon black along the thickness of the a cellulon paper. The evaporative layers made from lignocellulose aerogel and fiberglass film were prepared in a similar impregnating and drying method. The PVA-based evaporative layer was prepared by adding carbon-black slurry into the mixture of PVA and cross-linking agent before gelation happened.

**Fabrication of the SSEG**. The hydroscopic layer and the evaporative layer were pressed together at a pressure of 0.5 MPa with the plasma-treated side of the hydroscopic layer and high carbon-black-loaded side of the evaporative layer facing out. The edges of the as-pressed film were then encapsulated by parafilm. Two pieces of conductive carbon tape, each with a wide length of 0.5 cm, were connected to each side of the film to function as electrodes. For the fabrication of gold electrodes, the plasma sputtering method was adapted.

**Characterization and electric measurement**. X-ray photoelectron spectroscopy (XPS) was performed using an Escalab250Xi photoelectron spectrometer with Al Kα (1486.6 eV). Energy-dispersive X-ray spectra were performed using an Octane Elect EDS system. The zeta potential of each sample was measured using a Brookhaven NanoBrook zeta potential analyzer. The open-circuit voltage and short-circuit current of the device were measured using an electrometer of Keithley6500. The galvanostatic self-charge/discharge curves were tested using a PARSTST 3000A-DX with charge current and discharge current of 0 and 9 μA, respectively. The electrochemical impedance spectra of the SSEG were performed in a frequency range 1–100 kHz, with 5 mV a.c. amplitude. The transmittance and

reflectance spectra of the evaporative layer were measured in the range of 250-800 nm with a TU-1901 spectrophotometer attached to an integrating sphere. The light adsorption was calculated by $A = 1 - R - T$, $R$, and $T$ are the reflection and transmission, respectively. The temperature and degree of humidity in the atmosphere were recorded with a TH22R-EX Temperature Humidity Meter.

## Data availability
All data needed to evaluate the conclusions in the paper are present in the main text or the Supplementary Information. Additional data related to this paper are available from the corresponding author upon request.

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

## Acknowledgements
This work was supported by the National Key Research and Development Program of China (2019YFA0705400), Natural Science Foundation of Jiangsu Province (BK20212008), the Research Fund of State Key Laboratory of Mechanics and Control of Mechanical Structures (MCMS-I-0421K01), National Natural Science Foundation of China (12150002, 12172176), the Fundamental Research Funds for the Central Universities (NJ2020003, NZ2020001), A Project Funded by the Priority Academic Program Development of Jiangsu Higher Education Institutions.

## Author contributions
W.G., J.T., and S.F. designed the device architecture and experiments. J.T., L.L., and X.W. performed the experiments and characterization. W.G. and J.T. analyzed the data. The manuscript was written by J.T., Z.Z., J.Y., and W.G., and W.G. conceived and supervised the project.

## Competing interests
The authors declare no competing interests.
