## [Peer Review File · Nature Communications]

Self-sustained electricity generator driven by the compatible integration of ambient moisture adsorption and evaporationREVIEWER COMMENTS

Reviewer #1 (Remarks to the Author):

In this manuscript, the authors have reported a very simple and novel approach for generation of electricity from ambient humidity through integration of moisture absorbing and evaporating layer. The results are very interesting and encouraging and should be published in nature communications. The results are exciting and will contribute significantly to the field of moisture induced energy harvesting. But few major queries need to be addressed before the manuscript can be recommended for publication.

(i) In this manuscript, role of cellulon paper is not clear. Does author can use any type of membrane for fabricating the device? As authors have mentioned that the cellulon membrane provides negative charged channels for water flow, but the relation between extents of polar functionalities present in membrane with output voltage need to be studied more. Authors also may use different type membrane to investigate the role of membrane functionality on power generation.

(ii) What are power density and energy density value of these devices?

(iii) Does the device behave like a capacitor?

(iii) Does both proton and lithium ion flow through the channel? Or only proton flow? What is the ionic conductivity value of this device at 50% RH?

(iv) How output voltage varies with thickness of the device?

(v) What is the role of carbon black in the evaporating layer? Is it chosen for its hydrophobic character?

Reviewer #2 (Remarks to the Author):

This manuscript has shown a novel way to generate sustainable electricity from ambient humidity and natural evaporation, the authors have developed a new device prototype for enhanced power generation from ambient humidity, in which water vapor from air was adsorbed by a LiCl-loaded cellulon paper, a carbon-black-loaded cellulon paper to promote water evaporation, and the interfaces connected between the two parts could thus generate electricity around 0.78 V and a current of around 7.5 μ A, both of which can be sustained for more than 10 days. The work is novel and with good potentials for applications in microelectronic devices. Though this manuscript is well presented, major revisions are needed to improve the quality of this paper.

1. The heat and mass transfer process need to be described, especially the evaluation on the mass flow between two parts (sorption and evaporation) via multilayer interfaces. In the experiment, it was shown that the humidity contents changes along the different layers, it would be good to describe this with heat & mass transfer process.

2. The information of absorptivity of the black coated paper for evaporation of water need to be added.

3. The authors may need to add some information if there are any salt content transfer between moisture adsorption part (deliquescence) to water evaporative part. If yes, this may have limitation time to generate electricity, and LiCl will be consumed and released at the evaporative part. If not, how to prevent the LiCl content flow?

4. The evaporation of water content is under one sun illumination, how to find the application cases if this small devices adsorbe air humidity at one side while another side is heated by sun illumination to generate water vapor. During one sun illumination heating process, how to prevent heat conduction to the other side which is used to adsorbe moisture from air?

5. Some up-to-date literature might be cited with this new concept air humidity is energy and also air humidity sorption for sweat cooling, such as
a). Digestion of Ambient Humidity for Energy Generation

Joule, Vol. 4, Issue 12, p2532–2536 Published online: November 5, 2020.

b). A Thermal Management Strategy for Electronic Devices Based on Moisture Sorption-Desorption Processes, Joule, Vol. 4, Issue 2, p435–447 Published online: January 22, 2020

Reviewer #3 (Remarks to the Author):

This manuscript reported a self-sustained electricity generator by adopting a bilayer integrating hygroscopic layer and an evaporative layer. This prototype uses both heterogeneous materials assembled from a LiCl-loaded cellulon paper to facilitate moisture adsorption and a carbon-black-loaded cellulon paper to promote water evaporation. Exposing such a centimeter-sized device to ambient humidity can produce voltages of around 0.78 V and a current of around 7.5 μA , both of which can be sustained for a long time. The enhanced electric output and durability are due to the continuous ionic water flow within the cellulon papers. This work is interesting and promising for sustainable electricity generation.

Specific comments:

- 1 The hygroscopic layer was fabricated by introducing LiCl into a cellulon paper slice. The hygroscopic LiCl component could be dissociated into Li^+ and Cl^- ions. Due to the evaporative layer without introducing LiCl, there also remains the ions concentration gradient of Li^+ . Whether Li^+ ions, similar to H^+ , could diffuse from hygroscopic layer to evaporative driven by ions concentration gradient? If applicable, the distribution of LiCl component should be reasonably characterized during the power generation. If the diffusion of Li^+ ions is indeed existence, whether Li^+ ions diffusion in this work also contributes for electric output? Furthermore, the authors may adopt non-ionic hygroscopic agents without ions dissociation as substitution of LiCl component.
2. Given the irreversibility of Li^+ ions diffusion, the cyclic stability of power generation in this work should be demonstrated.
3. There are no analysis and evidences to confirm that the energy comes from ambient humidity and natural evaporation. The authors should rigorous elaborate the energy source for generating electricity.
4. The energy conversion efficiency and output power density need to be evaluated.
5. The authors mentioned, "The maximum voltage output was obtained when transportation and back-migration of positive ions driven by persistent water flow and electrostatic repulsion achieved balance." The sentences should be checked. Did the current disappears when transportation and back-migration of positive ions achieved balance? When did the device provide the maximum voltage in Figure 1c?
6. Will the voltage and current output still maintain stably after 10 days?
7. Discuss the influence of different hygroscopic salts on electric output. The applicable conditions of the relationship between Debye length and ions sensitivity should be checked.

Changes made in the revision:

1. Results from additional experiment for the mass transfer between the hygroscopic layer and the evaporative layer were presented in the revised Fig. 3, and discussed in the manuscript.
2. Additional experimental results and calculations for the power density and energy density of the SSEG were presented in Supplementary Fig. 2 and Note 1 in the revised Supplementary Information.
3. The water absorptivity of the evaporative layer was added in Supplementary Fig. 8 in the revised Supplementary Information.
4. Additional experiment results for excluding salt crystallization at the evaporative layer were presented in Supplementary Fig. 9, and discussed in Supplementary Note 2 in the revised Supplementary Information.
5. The evolution of the SSEG surface temperature was added in Supplementary Fig. 10, and the heat transfer process between the two layers was discussed in Supplementary Note 3 in the revised Supplementary Information.
6. The voltage output of the SSEG under one-sun illumination was added in Supplementary Fig. 11 in the revised Supplementary Information.
7. Transportation of ions during the power generation was presented and discussed in Supplementary Fig. 14 and Note 4 in the revised Supplementary Information.
8. The electric outputs of the SSEG under different environment conditions were presented in Supplementary Fig. 15, the energy source was accordingly discussed in Supplementary Note 5 in the revised Supplementary Information.
9. The voltage output of the SSEGs with non-ionic hygroscopic agents was added in Supplementary Fig. 16 in the revised Supplementary Information.
10. Additional experiment results for electric outputs of controlled devices with different constructing materials and different plasma treating time were presented and discussed in Supplementary Fig. 19 and Note 6.
11. The effect of device thickness on electric outputs was added and discussed in Supplementary Fig. 20 and Note 7 in the revised Supplementary Information.
12. Cyclic stability of the SSEG was presented and discussed in Supplementary Fig. 21 and Note 7 in the revised Supplementary Information.

Detailed revisions and discussions were made according to the reviewers' comments through the whole manuscript.

The revisions have been highlighted in blue color in the revised manuscript.

Response to the reviewers

We thank the three reviewers for their insightful comments and constructive suggestions. Accordingly, we have conducted extensive experiments and analyses to clarify all the raised concerns and made major revision with the new results being incorporated. The changes made in the revision are listed in detail in the following the reviewers' listing. The corresponding revision is also highlighted in the text.

Responses to comments from Reviewer #1

General Comment: *In this manuscript, the authors have reported a very simple and novel approach for generation of electricity from ambient humidity through integration of moisture absorbing and evaporating layer. The results are very interesting and encouraging and should be published in nature communications. The results are exciting and will be contribute significantly to the field of moisture induced energy harvesting. But few major queries need to be addressed before the manuscript can be recommended for publication.*

Response: We thank the reviewer for the positive assessment of our work and the following insightful comments and helpful suggestions.

Comment 1: *In this manuscript, role of cellulon paper is not clear. Does author can use any type of membrane for fabricating the device? As authors have mentioned that the cellulon membrane provides negative charged channels for water flow, but the relation between extents of polar functionalities present in membrane with output voltage need to be studied more. Authors also may use different type membrane to investigate the role of membrane functionality on power generation.*

Response: Thanks for the helpful suggestions. We carried out a series of experiments as suggested to clarify the role of cellulon paper as constructing material. Three types of membranes (lignocellulose aerogel, polyvinyl alcohol hydrogel and fiberglass film) were used to replace the cellulon paper for fabricating the devices, and the power generation performance of the obtained devices was accordingly tested. As shown in Figs. R1a and 1b, controlled devices made from lignocellulose aerogel and polyvinyl alcohol (PVA) hydrogel, both with negatively charged channels and abundant hydroxy groups, yields similar and even higher electric output of 0.9 V, 5.5 μ A and 0.5 V, 4 μ A,

respectively. As the PVA hydrogel contains water inside, the corresponding evaporative layer has a reduced water content gradient between two layers initially, which leads to a slight decrease in the electricity. By contrast, controlled device made from fiberglass film, which feature highly porous structure without functional groups and negatively charged channels, yields much lower electric output of about 0.15 V, 20 nA (Fig. R1c). The significant difference in electric output indicates that the porous structure is not the only requirement for the constructing material, while oxygen-containing functional groups and the negatively charged channels that provide free protons and selectivity to the opposite ions, play a crucial role in electricity generation. To clarify the effect of functional group concentration, we further measured the voltage outputs of several devices in which the materials were plasma-treated for different durations. As shown in Fig. R1d, the device without additional treatment produces a voltage of ~ 0.5 V. Introducing functional groups by plasma treatment significantly raises the voltage to a maximum of ~ 0.7 V at a treating time of 5 minutes and then slightly decreases with further extending the treatment, suggesting a saturation of functional group. Based on the results presented here and in the manuscript, the role of cellulon paper could be summarized in three aspects: providing binding sites and channels, releasing protons when moisture is captured and enabling high selectivity to opposite ions.

Fig. R1 | Electric outputs of controlled devices with different constructing materials and different plasma treating time. Voltage and current output of the device made from (a) lignocellulose, (b) polyvinyl alcohol hydrogel and (c) fiberglass. (d) Voltage output of controlled devices with different plasma treating time. The environmental conditions are 25 °C, 60% RH.

Comment 2: *What are power density and energy density value of these devices?*

Response: Thanks for the question. We calculated the power density and energy density value of the generators. The maximum power density of the device at different humidity was calculated as:

$$P_{max} = V \times J,$$

where V and J are the open-circuit voltage and the short-circuit current density, respectively. The areal power density approaches 32.9 nW cm^{-2} and 0.7 μW cm^{-2} at 20% and 60% RH (25°C), respectively.

The energy density of the generators at different humidity was measured via galvanostatic discharge method and further calculated as:

$$E = I \times \int_0^t U,$$

where I , U , t are the discharge current, voltage and discharge time, respectively. The gravimetric energy density (E_m) and volumetric energy density (E_v) were calculated as:

$$E_m = \frac{E}{m}, \quad E_v = \frac{E}{Ad},$$

where m , A , d are the mass, area and thickness of bilayer heterogenous materials, respectively. A gravimetric energy density of $0.15 \text{ mW}\cdot\text{h}\cdot\text{kg}^{-1}$ and a volumetric energy density of $0.04 \text{ mW}\cdot\text{h}\cdot\text{L}^{-1}$ are obtained at $25 \text{ }^\circ\text{C}$, $60\% \text{ RH}$.

Fig. R2 | Galvanostatic discharge curve of the SSEG at $25 \text{ }^\circ\text{C}$, $60\% \text{ RH}$. The discharge current is $9 \text{ } \mu\text{A}$.

Comment 3: *Does the device behave like a capacitor?*

Response: It is a very interesting question. Our generators and the parallel plate capacity indeed hold similarities in terms of bilayer structure and charge distribution, but they are fundamentally different. The most significant difference is that the charging process of the generator relies on spontaneous adsorption and evaporation of moisture, whereas charging a capacity always needs an external power source. This is essential to distinguish the two kinds of devices as energy harvesting device and energy storage device, respectively. Furthermore, the carriers flow in the two devices in different ways. Accompanying the moisture adsorption and water evaporation process of the generator, mobile positive ions inside the device will diffuse from the hygroscopic layer to the evaporate layer driven by an ion concentration difference and water flow, resulting in formation of current in the external circuit. However, there is no ion diffusion inside the capacity during the operation. Instead, charge carriers stored on the positive and

negative plates flow and are neutralized when load is connected.

Comment 4: *Does both proton and lithium ion flow through the channel? Or only proton flow? What is the ionic conductivity value of this device at 50% RH?*

Response: When the SSEG was exposed to the air, lithium chloride contained in the hygroscopic layer absorbed moisture spontaneously and partially been dissociated into lithium and chlorine ions, thus forming water content gradient and ion concentration gradient between the two layers. Lithium ions and protons dissociated from functional groups will diffuse from the hygroscopic layer to the evaporative layer driven by the gradients. To verify the transfer process of lithium ions, x-ray photoelectron spectroscopy of the evaporative layer surface was measured. After the device has operated for 4 days, the Li_{1s} spectra becomes detectable as shown in Fig. R3a, indicating the presence of slow lithium ion flow from the hygroscopic layer to the evaporative layer.

An ionic conductivity of $1.5 \mu\text{S}/\text{cm}$ of the SSEG at 50% RH was estimated based on the electrochemical impedance spectroscopy of the device (Fig. R3b).

Fig. R3 | Characterizations of the SSEG. (a) Li_{1s} X-ray photoelectron spectra of the evaporative layer surface after the device has operated for 4 days at 25°C , 60% RH. (b) The electrochemical impedance of the SSEG at 50% RH.

Comment 5: *How output voltage varies with thickness of the device?*

Response: Thanks for the insightful question. The effect of thickness on the performance of the device was investigated as shown in Fig. R4. The voltage increased

from 0.24 V to a maximum of ~0.75 V when the thickness of the device increased from 0.5 to 2 mm and then slightly decreased with further increasing the thickness to 4 mm. This is because the device thickness will affect the ion diffusion and the distribution of water content gradient. A SSEG with a thickness of 2 mm yielded an optimum voltage and current output of 0.78 V and 7.5 μ A, respectively.

Fig. R4 | Voltage and current outputs of the SSEG with different thickness tested at 25 °C and 60% RH.

Comment 6: *What is the role of carbon black in the evaporating layer? Is it chosen for its hydrophobic character?*

Response: Thanks for the valuable question. We selected carbon black as the essential material loaded in the evaporating layer for two reasons. First, the functionalized carbon black has both prominent hydrophobicity and photothermal effect (Fig. 1b, 2f), which facilitate the evaporation of water from the evaporative layer. Second, when water flows through the channels contained in porous carbon black, electricity will additionally be produced (*Nat. Nanotech 2017, 12, 317-321*), which results in a higher electric output of the device.

Responses to comments from Reviewer #2

General Comment: *This manuscript has shown a novel way to generate sustainable electricity from ambient humidity and natural evaporation, the authors have developed a new device prototype for enhanced power generation from ambient humidity, in which water vapor from air was adsorbed by a LiCl-loaded cellulon paper, a carbon-black-loaded cellulon paper to promote water evaporation, and the interfaces connected between the two parts could thus generate electricity around 0.78 V and a current of around 7.5 μ A, both of which can be sustained for more than 10 days. The work is novel and with good potentials for applications in microelectronic devices. Though this manuscript is well presented, major revisions are needed to improve the quality of this paper.*

Response: We thank the reviewer for the positive evaluation of our manuscript and the valuable suggestions. We have conducted additional experiments as suggested and revised the manuscript accordingly.

Comment 1: *The heat and mass transfer process need to be described, especially the evaluation on the mass flow between two parts (sorption and evaporation) via multilayer interfaces. In the experiment, it was shown that the humidity contents changes along the different layers, it would be good to describe this with heat & mass transfer process.*

Response: Thanks for the helpful suggestion. Control experiments were carried out to evaluate the mass and heat transfer process as suggested. First, two assembled device (device 1, 2) and one unassembled device with separate hygroscopic layer and evaporative layer (baseline) were sufficiently exposed to the environment condition at 25 °C and 70% RH until reaching mass equilibria. The outside surface of the evaporative layer of device 1 was sealed by parafilm to prevent the evaporation process. Mass evolutions of the hygroscopic layer and the evaporative layer of each device were recorded simultaneously. As shown in Fig. R5a, the mass change of each hydroscopic layer rises rapidly in the initial stage. However, the rising rates of devices 1 and 2 obviously decline with time, compared to the baseline describing only the hygroscopic process without mass transfer. The hygroscopic layer mass changes of baseline, devices 1 and 2 ultimately reach a plateau of 0.77, 0.59 and 0.54 g/g, respectively. The reduced mass change rates and values of devices 1 and 2 can be attributed to the mass transfer from the hygroscopic layer to the evaporative layer, which is also confirmed by the

mass change curves of the evaporative layer. The mass change of the evaporative layer in devices 1 and 2 increase slightly for the first two hours, during which water is captured by the hygroscopic layer and diffuse inwards. When the water reaches the interface between two layers, the mass transfer from the hygroscopic layer to the evaporative layer starts, resulting in a rapid increase of mass change rate. Compared to the near-zero mass change of the separate evaporative layer (baseline), the mass change of evaporative layer in the devices 1 and 2 reach up to 0.21 and 0.16 g/g, respectively. The relatively lower mass change in the evaporative layer in the device 2 can be attributed to the natural evaporation of water into air. Furthermore, the water content gradient inside the device is also confirmed by the difference of mass change of the hygroscopic layer and the evaporative layer.

To evaluate the heat transfer process between the two layers, two assembled devices were placed in an environment of 25 °C and 60% RH, while illumination of one sun was applied to one of the devices. The surface temperature of each layer of the two devices was measured and recorded simultaneously. As shown in Fig. R5b, temperature of the hygroscopic layer surface in dark presents an obviously higher level than that of the evaporative layer surface during the first 20 minutes. The exothermic behavior caused by rapid deliquescence of lithium chloride in the hygroscopic layer is responsible for the temperature difference, which gradually diminishes and disappears because of heat transfer from the hygroscopic layer to the evaporative layer and air. The direction of heat transfer between the two layers under illumination is opposite. The surface temperature of the evaporative layer is much higher than that of the hygroscopic layer due to its photothermal characteristic, resulting in a persistent heat transfer from the evaporative layer to the hygroscopic layer.

Fig. R5 | Mass and heat transfer process between the hygroscopic layer and the evaporative layer. (a) Mass evolution of different part of devices at 25 °C, 70% RH. (b) Temperature evolution of the hygroscopic layer and the evaporative layer under one-sun illumination and dark conditions. The temperature and relative humidity of the environment are 25 °C and 60%.

Comment 2: *The information of absorptivity of the black coated paper for evaporation of water need to be added.*

Response: Thanks for the valuable suggestion. The water absorptivity of the evaporative layer, which represents the ratio of water transferred from the hygroscopic layer, was measured at different RH. As shown in Fig. R6a, the water absorptivity of the evaporative layer increases with increasing the relative humidity and reaches 23.7%, 29.3% and 31.8% at 60%, 70% and 80% RH, respectively. In addition, light absorptivity of the evaporative layer was measured with a spectrophotometer attached to an integrating sphere (Fig. R6b). A light absorption of 94% is accordingly obtained in the range of 250-800 nm.

Fig. R6 | Water absorptivity and light absorptivity of the evaporative layer. (a) Water absorptivity of the evaporative layer at 25 °C and different RH. (b) Experimental absorption spectra of the evaporative layer measured in the range of 250-800 nm.

Comment 3: *The authors may need to add some information if there are any salt content transfer between moisture adsorption part (deliquescence) to water evaporative part. If yes, this may have limitation time to generate electricity, and LiCl will be consumed and released at the evaporative part. If not, how to prevent the LiCl content flow?*

Response: Thanks for the helpful suggestion. Chlorine element distribution in the evaporative layer was tracked by energy dispersive spectrum (Fig. R7a) to evaluate the salt content transfer between the two parts. When SSEG adsorbs moisture from air environment (60 %, 25 °C) for 6 hours, a small content of chlorine ions diffuse from the hygroscopic layer to the evaporative layer and is mainly distributed on the inner side of the evaporative layer, indicating that the transfer of chlorine ions is rather retarded and negligible. The ion selectivity of the negatively charged cellulon channels is responsible for the restricted chlorine ions flow. The diffusion of lithium ions was tracked by x-ray photoelectron spectra as shown in Fig. R7b. The presence of lithium ions at the outside surface of the evaporative layer cannot be detected until the device has operated continuously for 4 days, which reflects the slight transfer of lithium chlorine. To further confirm the restricted salt transfer between the two parts, an assembled device was exposed to environment of 25 °C with 70% RH, and one-sun illumination was applied on a half of the evaporative layer. After 12 hours of persistent intense moisture adsorption and water evaporation, there is no obvious salt formation

on the surface of the whole evaporative layer, indicating that LiCl cannot be massively consumed and crystallized at the evaporative layer during operating process.

Fig. R7 | Ion transfer and salt formation in the evaporative layer after moisture adsorption and water evaporation. (a) Element mapping images of energy dispersive spectrum of chlorine in the evaporative layer after adsorbing moisture for 6 hours (25 °C, 60% RH). (b) Li_{1s} X-ray photoelectron spectra of the evaporative layer outside surface after device has operated for 4 days at 25 °C, 60% RH. (c) Photograph of the evaporative layer surface after adsorbing moisture and evaporating for 12 hours under one-sun illumination (25 °C, 70 % RH).

Comment 4: *The evaporation of water content is under one sun illumination, how to find the application cases if this small device adsorb air humidity at one side while another side is heated by sun illumination to generate water vapor. During one sun illumination heating process, how to prevent heat conduction to the other side which is used to adsorb moisture from air?*

Response: Actually, electric outputs of the SSEG presented in original manuscript were all measured without applying extra illumination. The device can work well at night or indoors, and is operable in a wide range of humidity and temperature (Fig. 4a, 4b). We further measured the produced voltage of the device under one-sun illumination. An enhanced voltage output of about 0.9 V was obtained as shown in Fig. R8, which may result from the enhanced evaporation by illumination. Thus, an application case of “smart window” which is made of the bilayer devices could be devised, in which the device is able to adsorb moisture from indoor environment and pump water out to outdoor. As the cellulon paper is a poor conductor of heat, a proper increase of device thickness could prevent the heat transfer to some extent.

Fig. R8 | Voltage output of the SSEG under one-sun illumination.

Comment 5: *Some up-to-date literature might be cited with this new concept air humidity is energy and also air humidity sorption for sweat cooling, such as*

a).Digestion of Ambient Humidity for Energy Generation

Joule, Vol. 4,Issue 12,p2532–2536 Published online: November 5, 2020.

b).A Thermal Management Strategy for Electronic Devices Based on Moisture Sorption-Desorption Processes, Joule, Vol. 4, Issue 2, p435–447 Published online: January 22, 2020.

Response: Thanks for the helpful suggestion. The relevant references have been cited and introduced in the revised manuscript.

Responses to comments from Reviewer #3:

General Comment: *This manuscript reported a self-sustained electricity generator by adopting a bilayer integrating hygroscopic layer and an evaporative layer. This prototype uses both heterogenous materials assembled from a LiCl-loaded cellulon paper to facilitate moisture adsorption and a carbon-black-loaded cellulon paper to promote water evaporation. Exposing such a centimeter-sized device to ambient humidity can produce voltages of around 0.78 V and a current of around 7.5 μ A, both of which can be sustained for a long time. The enhanced electric output and durability are due to the continuous ionic water flow within the cellulon papers. This work is interesting and promising for sustainable electricity generation.*

Response: We thank the reviewer for the positive assessment of our work and offering insightful comments and suggestions. We have conducted additional experiments for clarification and revised the manuscript accordingly.

Comment 1: *The hygroscopic layer was fabricated by introducing LiCl into a cellulon paper slice. The hygroscopic LiCl component could be dissociated into Li⁺ and Cl⁻ ions. Due to the evaporative layer without introducing LiCl, there also remains the ions concentration gradient of Li⁺. Whether Li⁺ ions, similar to H⁺, could diffuse from hygroscopic layer to evaporative driven by ions concentration gradient? If applicable, the distribution of LiCl component should be reasonably characterized during the power generation. If the diffusion of Li⁺ ions is indeed existence, whether Li⁺ ions diffusion in this work also contributes for electric output? Furthermore, the authors may adopt non-ionic hygroscopic agents without ions dissociation as substitution of LiCl component.*

Response: Thanks for the insightful comment and helpful suggestion. When the SSEG was exposed to air, lithium chloride in the hygroscopic layer absorbed moisture spontaneously and is partially dissociated into lithium and chlorine ions, thus forming water content gradient and ion concentration gradient between the hygroscopic layer and the evaporative layer. Lithium ions and protons dissociated from functional groups, will diffuse from the hygroscopic layer to the evaporative layer driven by the water flow and ion concentration gradient. To verify the transfer process of lithium ions, x-ray photoelectron spectroscopy of the evaporative layer outside surface was measured as shown in Fig. R9a. The Li_{1s} spectra is detected after the device has operated for 4 days, before when no obvious characteristic peak of Li_{1s} can be detected, indicating that the lithium ions will slightly flow from the hygroscopic layer to the evaporative layer.

Furthermore, the lithium content of 3.75% is much higher than the chlorine content of 0.3% (Fig. R9b), which coherently imply that positive ions dominate the total diffused ions. Energy dispersive spectrum of chlorine element in the evaporative layer was further tracked to reveal the distribution of lithium chloride in electricity generation. After the device continuously operates for 6 hours, a small amount of chlorine ions diffuse from the hygroscopic layer to the evaporative layer, mainly distributed on the inner side of the evaporative layer (Fig. R9c), indicating that the transfer of chloride ions is rather retarded and negligible. The ion selectivity of the negatively charged cellulon channels is reasonable for the restricted chlorine ions flow. To further clarify whether lithium ions diffusion contributes to the electricity generation, three kinds of non-ionic hygroscopic agents (metal-organic framework of UIO-66, zeolite, clay) were prepared and applied to replace the lithium chloride component in the hygroscopic layer. As shown in Fig. R9d, SSEG with non-ionic hygroscopic agents of UIO-66, zeolite and clay generate voltage outputs of about 0.4, 0.25 and 0.2 V, respectively, lower than 0.78 V from that containing lithium chloride. Note that this difference in output voltage partly results from the higher hygroscopic flux of the LiCl-contained SSEG than other SSEGs. These results suggest that the lithium diffusion may contribute to electricity generation, but at a limited level.

Fig. R9 | Transportation of ions during the power generation and voltage output of the SSEG with non-ionic hygroscopic agents. (a) Li_{1s} X-ray photoelectron spectra and (b) elemental composition of the outside surface of the evaporative layer after the device has generated electricity stably for 4 days. (c) Element mapping images of energy dispersive spectrum of chlorine in the evaporative layer after adsorbing moisture for 6 hours. (d) Voltage output of the SSEGs with non-ionic hygroscopic agents. The experiments were carried out at 25 °C and 60 % RH.

Comment 2: *Given the irreversibility of Li^+ ions diffusion, the cyclic stability of power generation in this work should be demonstrated.*

Response: To evaluate the cyclic stability of electricity generation, a SSEG was connected to an external circuit with the switch alternately turned off to autonomously charge and turned on to discharge the device in a 1.5 min interval. As shown in Fig. R10a, the voltage of the device is resumed spontaneously after the discharging process and maintains a retention of 85% after 150 cycles. The high cyclic stability of the device could be attributed to the strong moisture capture ability of the hygroscopic layer and the directed water flow driven by the evaporation process. The cyclic stability of

electricity performance for a SSEG with periodic water adsorption-dehydration is also tested and demonstrated in Fig R10b. The generated electricity of the device decays slowly with increasing the cyclic number due to the surface passivation (*ACS Mater. Lett.* 2020, 2, 671-684) of LiCl during the water adsorption-dehydration process, but it remains a retention of 88% and 67% after 10 and 20 cycles, respectively.

Fig. R10 | Cyclic stability of the SSEG. (a) Cyclic performance of periodic autonomous charge-positive discharge for a SSEG. The testing interval of each charge-discharge cycle is 1.5 minutes. (b) Cyclic performance of electricity generation for a SSEG with periodic water adsorption-dehydration. The device was exposed to air until stably generating electricity for 2 hours and then dried at 100 °C for 1 hour to completely remove water. The tests were carried out at room temperature with a relative humidity of 40%.

Comment 3: *There are no analysis and evidences to confirm that the energy comes from ambient humidity and natural evaporation. The authors should rigorous elaborate the energy source for generating electricity.*

Response: Thanks for the helpful suggestion. We performed additional experiments and provided more detailed explanations to clarify the energy source for electricity

generation. In our original manuscript, it has been proven that electrochemical reaction between samples and electrodes or oxygen contained in air can hardly contribute to the electricity generation. To further clarify where the energy comes from, electricity generation performance of the SSEG was measured in different environment conditions as shown in Fig. R11. SSEGs maintain high electric outputs without any illumination or electromagnetic radiation input at 60% RH, suggesting that neither light nor electromagnetic radiation can be the energy source. The contribution of thermoelectric was further excluded by the near-zero electricity generation from the device with one-side heated at 0% RH. Only when the SSEG was exposed to the air with both moisture adsorption and water evaporation, electricity could be generated rapidly and sustained for a long term (Fig. R11). Otherwise, the electric output is significantly degraded (Figs. 3a-c and R11), indicating that electricity generation is closely related to the moisture adsorption and water evaporation through the device. As other energy sources have been strictly ruled out, the chemical potential changes of water during the moisture adsorption process (*Nat. Nanotech.* 2021, 16, 811-819, *Energy Environ. Sci.* 2019, 12, 1848-1856) and the ambient thermal energy absorbed by natural evaporation process (*Nat. Nanotech.* 2017, 12, 317-321) are reasonably considered as the main energy source.

Fig. R11 | Electricity generation performance of the SSEG under different environment conditions. (a) Voltage and (b) current output of the SSEG under different environment conditions. A heat plate of 50 °C was attached to the surface of hygroscopic layer to evaluate the contribution of thermoelectric effect between the two layers. Other

tests were carried out at room temperature.

Comment 4: *The energy conversion efficiency and output power density need to be evaluated.*

Response: Thanks for the worthy suggestion. The energy conversion efficiency and power output density of the device are evaluated as suggested. In the electricity generating process of the SSEG, the chemical potential energy changes of water from gaseous water to confined water during moisture adsorption process, and the ambient thermal energy adsorbed by water during the natural evaporation process are considered as the main energy source. The input energy is thus estimated in two parts. For the chemical potential energy changes of water, it can be expressed as:

$$\Delta G = \mu_g - \mu_c \quad (1),$$

where μ_g and μ_c represents chemical potential energy of gaseous water in air and confined water in device, respectively. Water adsorption is assumed to be an isothermal and isobaric process. According to the thermodynamic law, chemical potential energy can be expressed as:

$$\mu = \left(\frac{\partial G}{\partial n}\right)_{T,p} = \mu^\theta(T, p) + RT \ln \frac{p}{p^\theta} \quad (2),$$

where G , n , T , p , μ^θ , R and p^θ represents Gibbs free energy, mole numbers, temperature, pressure, standard chemical potential, ideal gas constant and standard pressure, respectively. Eq. (1) yields

$$\Delta G \approx RT \ln \frac{c_0}{c_0 - \Delta c} \quad (3),$$

where c_0 and Δc represents the concentration of water in atmospheric and the concentration variation of water. The maximum chemical potential energy change of water could be estimated to be 0.128 J at 298 K according to previous report (*Nat. Nanotech. 2021, 16, 811-819, Nature, 2020, 578, 550-554, Energy Environ. Sci. 2016, 9, 912-916*), which could be regarded as the energy input in the moisture adsorption process.

As the evaporation is occurred automatically without any artificial energy input (doing mechanical work, lighting, heating, et al.), and we take some electricity from

environmental heat that is hard to be used, it is not suitable to estimate the energy conversion efficiency by considering heat energy absorbed by water as the energy input term. Even so, we tried to estimate the energy conversion efficiency as a reference. For the ambient thermal energy adsorbed by water during natural evaporation process, it can be expressed as:

$$H = mr \quad (4),$$

where m and r represent the mass of evaporated water and heat of vaporization of water at 298 K, respectively. The natural evaporation is assumed to be an isothermal and isobaric process. The mass of evaporated water could be estimated by measuring the evaporative performance of the device in Fig. R5. The ambient thermal energy adsorbed by water is estimated to be 48.39 J at 298 K, which could be regarded as the energy input in the water evaporation process.

The output energy could be estimated by the generated electricity of the device and expressed as:

$$E_{output} = \int U(t)I(t)dt \quad (5)$$

where U , I and t represent the voltage output, current output and time of outputting electricity, respectively. The maximum output energy of 0.021 J for an hour is calculated based on the electricity generation performance of the device. Therefore, the energy conversion efficiency [$E_{output}/(\Delta\mu+H)$] of 0.4‰ could be estimated as a reference.

The maximum power density of the device at different humidity was calculated according to the following formula:

$$P = V \times J \quad (6)$$

where V and J are the open-circuit voltage and the short-circuit current density, respectively. The areal power density of the generators approaches 32.9 nW cm⁻² and 0.7 μW cm⁻² at 20% and 60% RH (25°C), respectively.

Comment 5: *The authors mentioned, “The maximum voltage output was obtained when transportation and back-migration of positive ions driven by persistent water flow and electrostatic repulsion achieved balance.” The sentences should be checked. Did the current disappears when transportation and back-migration of positive ions achieved balance? When did the device provide the maximum voltage in Figure 1c?*

Response: Thanks for your careful review and the helpful suggestion. We accordingly reconsidered the generation process of the maximum voltage and rephrased the sentence in the revised manuscript. Before moisture adsorption occurs, there are no water or free ions inside the SSEG, hence voltage or current output cannot be measured. When the device is exposed to ambient environment, lithium chloride contained in the hygroscopic layer rapidly absorbs moisture and is partly dissociated into Li^+ and Cl^- . Mobile Li^+ , Cl^- and protons dissociated from the surface functional groups will gradually construct the ion concentration difference between the two layers. Benefiting from the continuous vapor exhaling from the evaporative layer, moisture adsorption at the hygroscopic layer can sustain for a long term to maintain the water flow and ion concentration difference. The continuous voltage generated by the device in Fig. 1c is obtained at this stage. The water content gradient and ion concentration difference sustained by compatible integration of moisture adsorption and evaporation could induce a persistent driving force of water flow and ion transportation beyond the back-migration driven by electrostatic repulsion, thus resulting in a continuous current output for a long term.

Comment 6: *Will the voltage and current output still maintain stably after 10 days?*

Response: The measured voltage and current of the generator for more than 10 days were shown in Fig. R12. A continuous electric output of about 0.8 V and 8 μA for a duration of 300 hours is obtained, no obvious degradation of electricity generation performance could be detected after 10 days. As the moisture adsorption ability of the hygroscopic layer may reduce due to the surface passivation of lithium chloride after a long term of water adsorption-dehydration, electric outputs may thus decay at some point after a longer operating time.

Fig. R12 | Electric outputs of the SSEG for more than 10 days in an ambient environment (25 ± 5 °C, $60 \pm 10\%$ RH).

Comment 7: *Discuss the influence of different hygroscopic salts on electric output. The applicable conditions of the relationship between Debye length and ions sensitivity should be checked.*

Response: Thanks for the helpful suggestion. Four common hygroscopic salts including MgCl_2 , CaCl_2 , AlCl_3 and FeCl_3 were separately introduced into hygroscopic layer with an identical concentration. Electric outputs of the obtained devices were then measured successively. As the ion diffusion makes contribution to electricity generation, factors including ionic radius and ionic valence that can affect ion diffusion behavior will also affect the electric output of the device. As shown in Fig. 4d, the SSEG with LiCl as the hygroscopic agent outputs the maximum electricity, contributed to the effective ion diffusion due to small ionic radius and the low electrostatic repulsive force due to low ionic valence. Other hygroscopic agents with larger ionic radius or higher ionic valence result in reduced generation performance of the SSEG.

The statement on the relationship between Debye length and ion sensitivity has been deleted.

REVIEWER COMMENTS

Reviewer #1 (Remarks to the Author):

The authors of the Manuscript#: NCOMMS-21-29360A has carefully addressed all the comment raised by the reviewer 1 and the responses are satisfactory. The main manuscript is also modified accordingly. The results are very interesting and hope will contribute significantly to the field of moisture induced energy harvesting. The manuscript in the present form may be accepted for publication in Nature communication.

Reviewer #2 (Remarks to the Author):

The authors have well responded my concerns and suggestions, especially for the heat and mass transfer, and also the long term uses of salt content in this hydrogel. Some literature have been updated.

Now this article is well presented, it can be accepted as is.

Reviewer #3 (Remarks to the Author):

The authors have largely improved the manuscript according to the reviews' comments. It is acceptable after considering several small concerns about mechanism below.

1. If possible, the dynamic ion distribution changing is recommended to be traced, such as the ion distribution evolution of Li and Cl.
2. How much voltage and current are contributed correspondingly by each layer? If a porous electrode is inserted between the adsorption and desorption layer, it will help the understand of the whole process.
3. For the mechanism of the process, it seems the water flow is the key influence factor, any further enhancement of the performance can be done by offering mechanical water pressure?

Reviewer #4 (Remarks to the Author):

Tan et al. reported a self-sustained electricity generator that extracts energy from ambient moisture using a double-layered membrane. The membrane consists of a hygroscopic layer facilitated by salt incorporation and the other evaporative layer enabled by photothermal carbon materials. A sustained electric output is achieved by exposing the membrane to ambient humidity caused by the continual absorption and evaporation of moisture to create a water content gradient. The generated energy can be used to power some commercial electronics by combining multiple generators and energy storage devices. The concept is plausibly interesting, while the overall output performance does not appear to be as impressive as earlier reported water-enabled generators. The underlying working mechanism of SSEG is obscure, with few direct proofs and lengthy discussions. Some essential concerns should be explicitly depicted for further consideration.

1. The authors claim that the output performance of SSEG is higher, while the fact that its calculated gravimetric energy density of the SSEG is only 0.15 mW h kg⁻¹, tens of times lower than previous graphene oxide-based devices. Furthermore, the energy conversion efficiency is estimated to be 0.5%, which is significantly lower than in earlier studies. So, what is the work's significant improvement? The performance of the current SSEG should be compared to prior works.
2. The working mechanism for SSEG is unclear at the present stage. The authors attribute the power generation of SSEG to a combination of directional ions movement and sustained water evaporation, without additional direct evidence to back this up. If ions transport determines the power generation of SSEG, continuous electricity generation lasting more than 10 days means Li+

ions will migrate from the hygroscopic layer to the evaporative layer over 10 days, which is counterintuitive and unreasonable.

3. The interface in the membrane should hinder the transport of Li^+ ions and water from the hygroscopic layer to the evaporative layer. What about the evaporative layer's ionic conductivity? And how about the water content map was characterized, and what is the resolution to quantify water content?

4. What about the performance of SSEG in a closed system with no water exchange with the outside environment? Is the electrical signal declined to zero with less time?

5. The weight variation of SSEG during the electricity generation process should be recorded. And what are the wicking and evaporative rates during the device operation?

6. How long can the integration of SSEGs charge a cell phone? More details about device integration and electronics powering should be provided.

NCOMMS-21-29360A

Changes made in the revision:

1. Additional experiments and characterizations have been conducted to clarify the mechanism during the revision and the new results have been incorporated into the revised manuscript and Supplementary Information as outlined in the following.
 - a) Voltage and current outputs contributed by different layers have been presented in in the revised Fig. 3e, and discussed in the revised manuscript.
 - b) The evolution of SSEG weight during the device operation has been presented in Supplementary Fig. 8 in the revised Supplementary Information, and discussed in the revised manuscript.
 - c) The voltage output of the SSEG in a closed system has been presented in Supplementary Fig. 14 in the revised Supplementary Information, and discussed in the revised manuscript.
 - d) The dynamic distribution and content changes of chlorine and lithium ions have been added in Supplementary Fig. 16 in the revised Supplementary Information, and discussed in the revised manuscript.
 - e) The galvanostatic discharge curve of the device made from cellulose aerogel has been added in the revised Supplementary Fig. 22, and the corresponding gravimetric energy density of the device has been calculated and discussed in the revised manuscript.
2. Minor revisions and rewordings have been made according to the reviewers' comments through the whole manuscript.

The revised text and discussions are highlighted in blue color (like here) in the revised manuscript.

General Comment: *The authors of the Manuscript#: NCOMMS-21-29360A has carefully addressed all the comment raised by the reviewer 1 and the responses are satisfactory. The main manuscript is also modified accordingly. The results are very interesting and hope will contribute significantly to the field of moisture induced energy harvesting. The manuscript in the present form may be accepted for publication in Nature communication.*

Response: We thank the reviewer for the effort in reviewing our manuscript and the recommendation of acceptance.

Response to comments from Reviewer #2:

General Comment: *The authors have well responded my concerns and suggestions, especially for the heat and mass transfer, and also the long term uses of salt content in this hydrogel. Some literature have been updated. Now this article is well presented, it can be accepted as is.*

Response: We thank the reviewer for the effort in reviewing our manuscript and the recommendation of acceptance.

Response to comments from Reviewer #3:

General Comment: *The authors have largely improved the manuscript according to the reviews' comments. It is acceptable after considering several small concerns about mechanism below.*

Response: We thank the reviewer for the positive evaluation of our revised manuscript and the following helpful suggestions. We have conducted additional experiments as suggested and revised the manuscript accordingly. We hope that the revision has made a satisfied improvement.

Comment 1: *If possible, the dynamic ion distribution changing is recommended to be traced, such as the ion distribution evolution of Li and Cl.*

Response: Thanks for the helpful suggestion. The dynamic distribution of chlorine in the evaporative layer has been traced during ten days by energy dispersive spectrum. As shown in Figure R1a, a small amount of chlorine ions diffuses from the hygroscopic layer to the evaporative layer after six hours operation of the device, and are mainly distributed on the inner side of the evaporative layer. In the next ten days, the chlorine ions gradually diffuse and distribute evenly across the evaporative layer, while the chlorine content is little changed. The results are consistent with the presumption that the transfer of chlorine ions is rather retarded and even negligible because of the ion selectivity of the negatively charged cellulon channels.

Since lithium is too light to be detected by the energy dispersive spectrum, x-ray photoelectron spectroscopy of the external surface of the evaporative layer was measured to reveal the dynamic transfer of lithium ions (Figure R1b). The lithium content ratio within the external surface of the evaporative layer increases steadily from 0% to 6.68% during 10 days, indicating that lithium ions slowly flow from the hygroscopic layer to the evaporative layer.

Fig. R1 | Dynamic distribution and content changes of chlorine and lithium ions. (a) Element mapping images of energy dispersive spectrum of chlorine in the evaporative layer during ten days. (b) Evolution of lithium content ratio within the outer surface of the evaporative layer during ten days.

Comment 2: *How much voltage and current are contributed correspondingly by each layer? If a porous electrode is inserted between the adsorption and desorption layer, it will help the understand of the whole process.*

Response: Thanks for the question. Voltage and current contributed by each layer were detected by inserting an extra porous carbon electrode between the two layers. As shown in Figure R2, voltages induced by the hygroscopic layer (V_{1-2}) and the evaporative layer (V_{2-3}) are 0.22 V and 0.51 V, respectively. A sum of V_{1-2} and V_{2-3} is equal to the measured V_{1-3} , which represents the voltage output of the entire device. The result indicates that both layers contribute to the total voltage but the evaporative layer contributes the major part, consistent with the speculation that the output power of SSEG results from water evaporation and ion movement. The current outputs of the hygroscopic layer, the evaporative layer and the entire device are 12.5 μA, 7.6 μA and 7.5 μA, respectively. The higher current value of the hygroscopic layer could be attributed to its lower internal resistance of ~16.2 kΩ. In contrary, the evaporative

layer with a higher internal resistance of $\sim 71.4 \text{ k}\Omega$ provides a lower current output.

Fig. R2 | Voltage and current outputs of different component layers at 25 °C and 60% RH.

Comment 3: *For the mechanism of the process, it seems the water flow is the key influence factor, any further enhancement of the performance can be done by offering mechanical water pressure?*

Response: It is a very interesting question. Since it is difficult to apply precise mechanical water pressure to the device in ambient environment, we try our best to verify the speculation by boosting the water flow in other ways. As shown in Figure R3, the power output performance of the device can be further enhanced by offering external driving force, such as applying illumination or wind to the evaporative layer, to promote evaporation and boost water flow. For the same principle, it can reasonably be inferred that applying mechanical water pressure can also boost the output power to a certain extent.

Fig. R3 | Voltage outputs of the SSEG under environmental conditions. The curves in red, blue and green represent the voltage of the device under one-sun illumination, wind (5 m/s) and no external stimulation, respectively.

Response to comments from Reviewer #4:

General comment: *Tan et al. reported a self-sustained electricity generator that extracts energy from ambient moisture using a double-layered membrane. The membrane consists of a hygroscopic layer facilitated by salt incorporation and the other evaporative layer enabled by photothermal carbon materials. A sustained electric output is achieved by exposing the membrane to ambient humidity caused by the continual absorption and evaporation of moisture to create a water content gradient. The generated energy can be used to power some commercial electronics by combining multiple generators and energy storage devices. The concept is plausibly interesting, while the overall output performance does not appear to be as impressive as earlier reported water-enabled generators. The underlying working mechanism of SSEG is obscure, with few direct proofs and lengthy discussions. Some essential concerns should be explicitly depicted for further consideration.*

Response: We thank the reviewer for reading our revised manuscript and providing helpful comments. In this revision, we have made our best to perform additional experiments and analyses to fully address the reviewer's concerns on the importance and the underlying mechanism of our work. The item-by-item responses to detailed comments are provided in the following.

Comment 1: *The authors claim that the output performance of SSEG is higher, while the fact that its calculated gravimetric energy density of the SSEG is only 0.15 mW h kg⁻¹, tens of times lower than previous graphene oxide-based devices. Furthermore, the energy conversion efficiency is estimated to be 0.5%, which is significantly lower than in earlier studies. So, what is the work's significant improvement? The performance of the current SSEG should be compared to prior works.*

Response: Thanks. Here we would like to strength the advances of our work over prior arts and the significance of this work. The moist-electric generators (MEGs) reported by Prof. Qu's group can produce pulse-type electrical output upon a change in the relative humidity (*Adv. Mater.*, **2015**, 27, 4351, *Nat. Commun.* **2018**, 9, 4166). However, these devices **cannot work sustainably** and require cyclical moisture feeding with quite a high gradient in humidity. The problem was alleviated in subsequent studies by engineering material composition and device structures to endow the devices with continuous capability of moisture adsorption (*Energy Environ. Sci.*, **2018**, 11, 2839; *Angew Chem. Int. Ed. Engl.*, **2016**, 55, 8003). For instance, a

moist-electric generator based on graphene oxide and sodium polyacrylate can produce continuous voltage output for more than 100 hours under ambient environment (with power density and gravimetric energy density of **0.07 $\mu\text{W}\cdot\text{cm}^{-2}$ and 0.25 $\text{mW}\cdot\text{h}\cdot\text{kg}^{-1}$** at 80% RH and 25 °C, and 5.23 $\text{mW}\cdot\text{h}\cdot\text{kg}^{-1}$ at 45 °C, *Energy Environ. Sci.*, **2019**, 12, 1848, possibly the work the reviewer mentioned). Similar progress was achieved in polyelectrolyte films and biological nanowires (*Nat. Nanotechnol.*, **2021**, 16, 811; *Nature*, **2020**, 578, 550). However, the electrical output of these devices, **especially the current output**, always exhibits a **rapid decay** (Figure R4a-c), which heavily limits their applications. One of the key factors is that the devices will finally reach a saturated state of moisture adsorption, thereby eliminating the water gradient inside the device. By contrast, the devices of evaporation-induced electricity can produce sustainable electricity but need water source. **In this work, we manage to solve this problem with this new prototype, which could maintain the water content gradient inside the device by integrating moisture adsorption with water evaporation.** The device can spontaneously produce a continuous voltage output of ~0.8 V and a continuous current output of ~8 μA , both of which can last for more than 300 hours **without notable decay** (Figure R4d). This was never achieved before, so we used “enhanced” to describe the device performance in the manuscript. More importantly, we have extended the proposed prototype to different constructing materials as well as non-ionic hygroscopic agents, providing a competitive strategy for sustained power generation of MEGs.

The persistence of the electric output is what we have been focusing on, and the gravimetric energy density of SSEG can be raised easily by replace cellulon paper with other low density constructing materials. When we replace it by cellulose aerogel with a low density and abundant hydroxyl, the power density and gravimetric energy density of the SSEG can be raised to **1.4 $\mu\text{W}\cdot\text{cm}^{-2}$ and 4.7 $\text{mW}\cdot\text{h}\cdot\text{kg}^{-1}$** at **55% RH and 23 °C**, respectively (Figure R4e). Since ambient thermal energy absorbed by evaporation was considered as the energy input, the energy conversion efficiency of the device is low. However, since the device can run automatically without any artificial energy input (e.g. mechanical work, lighting, heating, et al.), it is unfair to

take this energy conversion efficiency as the criterion to evaluate the device performance.

We have carefully revised the manuscript accordingly.

Fig. R4 | Electric outputs of different moist-electric generators (MEGs) and the

Editorial Note: Panel A above republished with permission of the Royal Society of Chemistry, from All-region-applicable, continuous power supply of graphene oxide composite, Huang et al., *Energy Environ. Sci.*, **12**, 1848-1856 (2019); permission conveyed through Copyright Clearance Center, Inc. Panel B above reprinted by permission from Springer Nature Customer Service Centre GmbH: Springer Nature, Wang, H., Sun, Y., He, T. et al. Bilyer of polyelectrolyte films for spontaneous power generation in air up to an integrated 1,000 V output. *Nat. Nanotechnol.* **16**, 811-819 (2021). Panel C above reprinted by permission from Springer Nature Customer Service Centre GmbH: Springer Nature, Liu, X., Gao, H., Ward, J.E. et al. Power generation from ambient humidity using protein nanowires. *Nature* **578**, 550-554 (2020).

SSEG. Electric outputs of MEG based on (a) graphene oxide and sodium polyacrylate (*Energy Environ. Sci.*, **2019**, 12, 1848), (b) heterogeneous polyelectrolyte (*Nat. Nanotechnol.*, **2021**, 16, 811), (c) and protein nanowires (*Nature*, **2020**, 578, 550). (d) Electric outputs of the SSEG for more than 300 hours in an ambient environment (25 ± 5 °C, $60 \pm 10\%$ RH). (e) Voltage and current outputs (f) and galvanostatic discharge curve of the device made from cellulose aerogel. The environmental conditions are 23 °C, 55% RH. The discharge current is 10 μ A.

Comment 2: *The working mechanism for SSEG is unclear at the present stage. The authors attribute the power generation of SSEG to a combination of directional ions movement and sustained water evaporation, without additional direct evidence to back this up. If ions transport determines the power generation of SSEG, continuous electricity generation lasting more than 10 days means Li⁺ ions will migrate from the hygroscopic layer to the evaporative layer over 10 days, which is counterintuitive and unreasonable.*

Response: Thanks. To address the reviewer's concern about the proposed working mechanism, additional experiments on evolution of lithium content within the evaporative layer and respective electric output contribution of different layers, were conducted to provide more direct evidence to clarify the working mechanism. In the last revision, we have found that SSEGs with non-ionic hygroscopic agents can produce voltage output of ~0.4 V, which can be a piece of direct evidence for the contribution of evaporation-induced electricity (no contribution of lithium ions). The transfer of lithium ions was further verified by XPS measurement with ionic hygroscopic agents. Based on these results, we inferred that the electric output of the devices comes from evaporation-induced electricity and additional ion movement. We understand the reviewer's concern about the continuous lithium ions transport in a long period. In this revision, x-ray photoelectron spectroscopy of the external surface of the evaporative layer during ten days was measured to reveal the dynamic transfer of lithium ions. As shown in Figure R5a, the lithium content ratio within the external surface of the evaporative layer presents a trend of continuous increase from 0% to 6.82% during ten days' operation of the device, which suggests that lithium ions do diffuse continuously from the hygroscopic layer to the evaporative layer. The

interface between the two layers and the relatively hydrophobicity of the evaporative layer can be the reasons for the slow transfer. Furthermore, the different electric output contributions of the two layers provide another piece of evidence for the working mechanism. As shown in Figure R5b, the hygroscopic layer with only ion movement (protons and lithium ions) yields a voltage of 0.22 V, while the evaporative layer with both evaporation and ion movement yields an obviously higher voltage of 0.51 V. The result indicates that the sustained water evaporation and proton transfer play a more important role in electricity generation. We have revised the manuscript accordingly by adding new experimental results and discussion.

Figure R5 | Dynamic transfer of lithium ions and electric contributions of different layers. (a) Evolution of lithium content ratio on the external surface of the evaporative layer. (b) Voltage and current outputs of different layers at 25 °C, 60% RH.

Comment 3: *The interface in the membrane should hinder the transport of Li⁺ ions and water from the hygroscopic layer to the evaporative layer. What about the evaporative layer's ionic conductivity? And how about the water content map was characterized, and what is the resolution to quantify water content?*

Response: Yes, the interface will hinder the transport of ions and water, resulting in a significant difference in ionic conductivity between the hygroscopic layer and the evaporative layer. At 40% relative humidity, the ionic conductivity of the evaporative layer is 1.2 μS/cm, while that of the hygroscopic layer is 93 μS/cm (calculated based on electrochemical impedance spectra of the device in a frequency of 100kHz, with 5

mV a.c. amplitude). The water content map was characterized by gravimetrically analyzing water absorbed by each unit. Specifically, the bilayer structure was divided into seven units (four in the hygroscopic layer and three in the evaporative layer) after sufficiently exposed to ambient environment at 60% RH. Each unit was weighed with an analytical balance and then transferred into an oven for complete dehydration. The dehydrated units were weighed again and the water contained in each unit was thus calculated to depict the water content map. The resolution corresponds to the film thickness, which is 0.25 mm for the hygroscopic layer, and 0.33 mm for the evaporative layer.

Comment 4: *What about the performance of SSEG in a closed system with no water exchange with the outside environment? Is the electrical signal declined to zero with less time?*

Response: As shown in Figure R6, a new SSEG in a closed system without any moisture adsorption or evaporation cannot produce any electric output, while a device in an open system yields a voltage of about 0.6 V. When a normally operating device is transferred into a closed system, the voltage declines to near zero in 2.5 hours.

Figure R6 | Voltage output of the SSEG in a closed system. The device was wrapped completely with parafilm in a closed system.

Comment 5: *The weight variation of SSEG during the electricity generation process should be recorded. And what are the wicking and evaporative rates during the device operation?*

Response: Thanks for your helpful suggestion. The evolution of SSEG weight was

traced during the device operation as suggested. As shown in Figure R7, the weight of the device increases rapidly and then reaches a plateau of about 130% the initial weight. In the plateau, the moisture adsorption rate and the evaporation rate gradually reach an equilibrium of $39 \text{ g}\cdot\text{m}^{-2}\cdot\text{h}^{-1}$, which was calculated based on the mass change in the evaporative layers of the device with/without water evaporation.

Figure R7 | Weight variation of SSEG and the evaporative layer. (a) Evolution of SSEG weight during the device operation. (b) Mass change in the evaporative layer of the devices with/without water evaporation. The environmental conditions are $25 \text{ }^\circ\text{C}$, 70% RH.

Comment 6: *How long can the integration of SSEGs charge a cell phone? More details about device integration and electronics powering should be provided.*

Response: Thanks for suggestion. In the demo experiments, four capacitors with 1000 microfarads were charged for 10 minutes by the integrated SSEGs. Then the charged capacitors were used to charge a cell phone and a charging time of 15 seconds was obtained. By contrast, the integrated heterogeneous moist-electric generators need a charging time of over ten hours to charge a capacity of 330 microfarads, which in turn was demonstrated to light a bulb (*Nat. Nanotechnol.*, **2021**, 16, 811). More importantly, the demo experiments show that the direct current generated by SSEGs can be stored easily with no need of extra rectifiers and power management circuit, which greatly reduce the energy consumption compared to devices with unstable electric output. We have added these details in the revised manuscript.

REVIEWERS' COMMENTS

Reviewer #3 (Remarks to the Author):

The revision is satisfactory for acceptance.

Reviewer #4 (Remarks to the Author):

The authors have well addressed the referee's concerns and suggestions. Additional experiments have been conducted to explain the working mechanism, and the results are clear to support their conclusions. The manuscript has been well revised, and it can be accepted now.

Response to comments from Reviewer #3:

General Comment: *The revision is satisfactory for acceptance.*

Response: We thank the reviewer for the effort in reviewing our manuscript and the recommendation of acceptance.

Response to comments from Reviewer #4:

General Comment: *The authors have well addressed the referee's concerns and suggestions. Additional experiments have been conducted to explain the working mechanism, and the results are clear to support their conclusions. The manuscript has been well revised, and it can be accepted now.*

Response: We thank the reviewer for the effort in reviewing our manuscript and the recommendation of acceptance.